# Authentication and Market Survey of Sweet Birch (*Betula lenta* L.) Essential Oil

**DOI:** 10.3390/plants11162132

**Published:** 2022-08-16

**Authors:** Noura S. Dosoky, Ambika Poudel, Prabodh Satyal

**Affiliations:** Essential Oil Science, dōTERRA International, 1248 W 700 S, Pleasant Grove, UT 84062, USA

**Keywords:** sweet birch, *Betula lenta*, essential oil, *o*-guaiacol, veratrole, 2-*E*-4-*Z*-decadienal, 2-*E*-4-*E*-decadienal, dimethyl 2-hydroxyterephthalate

## Abstract

Sweet Birch (*Betula lenta*) has several economic and medicinal uses. Very little is known about the chemical composition of *B. lenta*. In this study, the volatile compositions of the bark of *B. lenta* from authentic and commercial sources were assessed by gas chromatography-mass spectrometry (GC–MS) and gas chromatography–flame ionization detection (GC–FID). Overall, more than 60 compounds were identified in natural sweet birch EO obtained by hydro-distillation. The oil was dominated by methyl salicylate (93.24–99.84%). A good approach to distinguishing wintergreen and birch oils would be biomarker-based analysis. The biomarkers are selected based upon three main criteria: (1) the marker should be commercially unavailable or too expensive which renders the adulteration process very costly, (2) The marker should be detected consistently in all the tested authentic EO samples, and (3) A birch EO marker should be found exclusively in birch EO, not in wintergreen and vice versa. The minor components *o*-guaiacol, veratrole, 2-*E*-4-*Z*-decadienal, and 2-*E*-4-*E*-decadienal were identified as natural marker compounds for authentic sweet birch oil. Surprisingly, none of the tested 27 commercial samples contained any of the identified birch markers. The detection of wintergreen markers such as vitispirane and β-dehydroelsholtzia ketone, the synthetic marker dimethyl-2-hydroxyterephthalate, and ricenalidic acid lactone suggest the addition of wintergreen, synthetic methyl salicylate, and castor oil, respectively. This is the first report to identify birch biomarkers to the best of our knowledge.

## 1. Introduction

*Betula* species (Betulaceae) are largely found in regions north of the equator. Commonly known as birch, they have been used for centuries in various economic and medicinal applications [1]. *Betula* species are beneficial in cases of inflammation, infections, stomachache, urinary tract disorders, and skin problems [1,2]. *Betula* species contain flavonoids, glycosides, phenolics, saponins, sterols, tannins, and terpenes [3]. Being a rich source of antimicrobial compounds supports their traditional use to treat infections [4]. All parts of the tree, especially the wood, are used as raw materials in the paper and furniture industries in addition to charcoal production, dietary supplements, and cosmetics [5,6].

*Betula lenta* L., commonly known as sweet birch, black birch, mahogany birch, cherry birch, and southern birch, is one of the dark-barked birch species. *B. lenta* is a deciduous tree native to the eastern portion of North America. It is frequently found in northern forest ecosystems in the Appalachian Mountains. In the USA, it is found in Connecticut, Georgia, Kentucky, Maine, Maryland, Massachusetts, New Hampshire, New Jersey, New York, North Carolina, Ohio, Pennsylvania, Rhode Island, South Carolina, Tennessee, Vermont, Virginia, and West Virginia [7]. The tree can reach up to 30 m tall. *B. lenta* is a pharmacologically important tree. Traditionally, infusions and teas made of *B. lenta* bark have been taken for stomachache and lung diseases, while teas from bark and twigs are beneficial for fevers. *B. lenta* EO is an anti-inflammatory agent and is suitable for skin problems, rheumatism, and bladder infections. The young sweet birch tree has smooth bark with distinctive horizontal lenticels like other birch species. The mature tree has rough dark brown bark with irregular vertical cracks [8]. Birch bark is one of the abundant, underutilized natural resources [9]. It is the primary waste product of birch processing (about 10–15% of birch mass) [5]. The outer bark of white-barked birches is a good source of pentacyclic triterpenoids especially lupeol and betulin [9]. Several triterpenoids were identified in the outer bark of *B. lenta,* including betulin, betulone, betulinic acid, betulin-3-caffeate, lupeol, lupenone, lup-20(29)-ene-3ß,30-diol, lup-20(29)-ene-3ß-ol-30-al, lup-20(29)-ene-3ß,28-diol-30-al, and lupan-3ß,20diol [9]. *Betula* essential oils are generally produced by hydro-distillation. The bark of *B. lenta* is a known source of methyl salicylate and ethyl salicylate [10,11]. A significant amount of attention has been given to the authentication of wintergreen (*Gaultheria* spp.) EO. The authenticity of the essential oils of *B. lenta* offered in the market must be regarded with a similar attention. Unlike birch EO, the authentication of wintergreen EO has been the subject of several studies since 1986 [12]. Wintergreen and sweet birch EOs have a very similar chemical composition, dominated by methyl salicylate (about 99%). To authenticate the naturalness of this ester, it is easy to detect by GC-MS, preferably in SIM mode, the impurities that accompany the synthetic products acting as markers of the synthetic origins [12,13,14,15]. The efficiency of this task is enhanced using ^13^C, and ^2^H isotopic ratio measurements [14,15], and quantitative ^2^H-ERETIC-NMR [16]. Ultimately, ^14^C dating helps to ensure the contemporary origin of methyl salicylate [14]. Due to the high cost of natural birch oil, commercial birch oils are heavily adulterated [17]. Birch oil can be adulterated by adding a cheaper oil or synthetic methyl salicylate to increase the profit or by completely replacing the natural methyl salicylate with the inexpensive synthetic option. Therefore, the current study explores the volatile composition of the EO extracted from the bark of *B. lenta* obtained from trusted sources from Pennsylvania, USA, and compares the composition to the oils available in the US market. To the best of our knowledge, the EO composition of the bark of *B. lenta* has not been previously reported. Moreover, the essential oils of *B. lenta* were examined to identify unique chemical markers to help with authentication or adulteration detection.

## 2. Materials and Methods

### 2.1. Essential Oils

*B. lenta* bark samples were collected from McKean County, Pennsylvania, USA. Sweet birch grows in this area along with red maple (*Acer rubrum* L.), sugar maple (*Acer saccharum* Marshall), black cherry (*Prunus serotina* Ehrh), American beech (*Fagus grandifolia* Ehrh), Eastern hemlock (*Tsuga canadensis* (L.) Carriere), and yellow birch (*Betula alleghaniensis* Britt). *B. lenta* essential oils from authentic sources (hydro-distilled in both lab and industrial settings) and commercial suppliers were acquired from the EO collection of the Aromatic Plant Research Center (APRC, Lehi, UT, USA). Samples A1-A18 were distilled in industrial distillers while A19-A21 were extracted in the laboratory. The distillation conditions have been optimized in our previous work on wintergreen essential oil [13]. Birch bark samples were air-dried and then fermented in water for 8–10 h at 50 °C prior to distillation. Fermentation was over when the pH reached 3–4 and held this number for more than two hours. In the lab, the plant material was distilled for 6–10 h in a Clevenger-type apparatus. In the industrial setting, the plant material was distilled for 6–10 h at 600 kPa in a 5.5 m^3^ distiller, starting slowly for 2 h and then stabilizing around a flow rate of 2 L/min. Twenty-seven commercial birch essential oil products were purchased online. The product labels of most of these samples were labeled as “100% Pure Essential oil” or “100% pure Therapeutic Grade Essential oil” (Table 1).

### 2.2. Gas Chromatography-Mass Spectrometry (GC–MS) Analysis 

Natural and commercial birch oil samples were analyzed using a gas chromatograph coupled to a mass spectrometer QP2010 Ultra (Shimadzu Scientific Instruments, Columbia, MD, USA) with electron impact (EI) mode with 70 eV. The separation of the analytes was carried out by using a ZB5MS GC capillary column, using 40–400 *m*/*z* range scans with a scan rate of 3.0 scan/s. The column temperature was set at 50 °C for 2 min and then increased at 2 °C/min to the temperature of 260 °C. The carrier gas was helium with a constant flow rate of 1.37 mL/min. The injector temperature was kept at 260 °C. For each essential oil sample, 1:10 *v*/*v* solution in dichloromethane (DCM) was prepared, and 0.3 μL was injected using a split ratio of 1:30. The essential oil components were identified by comparing mass spectral fragmentation patterns (over 80% similarity match) and retention indices (RI) based on a series of homologous C8–C20 n-alkanes with those reported in databases (NIST database, and our in-house library) using the Lab Solutions GCMS post-run analysis software version 4.45 (Shimadzu Scientific Instruments, Columbia, MD, USA) [18].

### 2.3. Gas Chromatography–Flame Ionization Detection (GC–FID) Analysis 

Analysis of natural birch essential oil was carried out using a Shimadzu GC 2010 equipped with a flame ionization detector (Shimadzu Scientific Instruments, Columbia, MD, USA), as previously described [18] with a ZB-5 capillary column (Phenomenex, Torrance, CA, USA).

## 3. Results and Discussion

### 3.1. Authentic B. lenta Oil

Sweet birch samples were hydro-distilled both in the lab and in industrial distillers by trusted sources. Thus, there is no question about authenticity. The EO yield was in the range of 0.07–0.1%. The odor can be described as papery, lightly weak wintergreen, sweet, and lightly woody. There were no significant differences between the lab-distilled oils vs. industrially distilled ones. The chemical compositions of authentic birch EO (21 samples) are summarized in Table 2. Overall, more than 60 compounds were identified in natural sweet birch EOs. The oils are dominated by methyl salicylate (93.24–99.84%). Methyl salicylate is an analgesic phenolic ester [19] that dominates wintergreen [17] and sweet birch [11] oils. During the hydro-distillation of wintergreen *(Gaultheria fragrantissima* Wall), the glycoside gaultherin is converted to methyl salicylate, which is the major compound of the oil (above 98%) [17]. Some variations were observed in some of the minor components. o-Guaiacol, veratrole, ethyl salicylate, 2-*E*-4-*Z*-decadienal, 2-*E*-4-*E*-decadienal, and methyl o-anisate were consistently detected in all the authentic samples. The composition of *B. lenta* bark in the current study is quite different from that of other birch bark oils. The EO obtained from *B. nigra* bark was dominated by fatty acids and fatty acid-derived compounds (51.2–80.4%) and saturated normal alkanes (4.5–29.8%) [2,20,21]. The inner bark of *B. pendula* yielded an EO made of methyl salicylic, palmic, phenic, and behenic acids, and sesquiterpenes [22]. In another study from New Zealand, the EO of the inner bark of *B. pendula* had *E*-α-bergamotene (31%) and α-santalene (19%) as the main components. In comparison, the major components of *B. papyrifera* inner bark oil were *E*-α-bergamotene (18%), *ar*-curcumene (12%), *E*-β-farnesene (12%), *Z*-β-farnesene (10%) and Z-α-bergamotene (8%) [23]. The main constituents of the EO obtained from the bark of *B. pubescens* from Russia were α-santalene (2.0%), *E*-α-bergamotene (3.5%), *E*-β-bergamotene (0.8%), α-epoxysantalene (0.3%), *E*-α-epoxybergamotene (0.3%) and *E*-β-epoxybergamotene (0.4%) [24]. 

Enantiomeric distribution is one of the popular methods of detecting adulteration in essential oils [25]. However, it was not possible to perform chiral analysis on *B. lenta* bark oil since methyl salicylate lacks a chiral core, and the minor chiral components are present in very low quantities. It is easy to confuse wintergreen and sweet birch oils due to their high methyl salicylate content. A good approach to distinguishing wintergreen and birch oils would be biomarker-based analysis. The biomarkers are selected based upon three main criteria: (1) the marker should be commercially unavailable or too expensive which renders the adulteration process very costly, (2) The marker should be detected consistently in all the tested authentic EO samples, and (3) A birch EO marker should be found exclusively in birch EO, not in wintergreen and vice versa. Table 3 shows a comparison between authentic sweet birch and wintergreen oil compositions. A total of 18 natural compounds are common in both oils. Ethyl salicylate and methyl o-anisate are common markers in both oils; however, they are present in higher amounts in sweet birch EO. Therefore, we can identify o-guaiacol, veratrole, 2-*E*-4-*Z*-decadienal, and 2-*E*-4-*E*-decadienal as natural marker compounds for authentic *B. lenta* oil (Figure 1).

### 3.2. Commercial Sweet Birch Oil

Twenty-seven commercial birch oil samples were obtained from commercial vendors available in the US market. The chemical compositions of commercial birch EOs are shown in Table 4, with more than 100 identified components. Similar to the natural EO, commercial sweet birch EOs were dominated by methyl salicylate (56.71–99.9%). Minor components varied greatly between commercial samples. Woods and colleagues analyzed a commercial EO sample that was exclusively made of methyl salicylate [2,20]. Sweet birch markers (o-guaiacol, veratrole, 2-*E*-4-*Z*-decadienal, and 2-*E*-4-*E*-decadienal) were not detected in any of the tested commercial samples. Ethyl salicylate was absent from 11 samples. Interestingly, wintergreen markers such as vitispirane (10 occurrences) and β-dehydroelsholtzia ketone (3 occurrences) were detected in the tested samples. This finding suggests that wintergreen oil is available in the market as birch EO or the addition of wintergreen oil to natural birch oil [13]. Moreover, synthetic markers such as dimethyl-2-hydroxyterephthalate (6 occurrences) were also detected, which suggests the addition of synthetic methyl salicylate [17]. Ricenalidic acid lactone, a marker of the addition of castor oil, was detected in one sample. 

## 4. Conclusions

Hydro-distilled *Betula lenta* L. essential oils showed almost similar chemical compositions, with methyl salicylate as the main component (93.24–99.84%). Four biomarkers, namely, o-guaiacol, veratrole, 2-*E*-4-*Z*-decadienal, and 2-*E*-4-*E*-decadienal were identified for the natural *B. lenta* oil. These markers can be used to distinguish between sweet birch and wintergreen oils and may be used in sweet birch oil authentication and adulteration detection. Interestingly, none of the tested commercial samples contained any of the identified birch EO markers. The detection of wintergreen markers such as vitispirane and β-dehydroelsholtzia ketone, the synthetic marker dimethyl-2-hydroxyterephthalate, and ricenalidic acid lactone suggest the addition of wintergreen, synthetic methyl salicylate, and castor oil, respectively. Further investigations on the evaluation of biological activities of *B. lenta* essential oil are required.

## Figures and Tables

**Figure 1 plants-11-02132-f001:**
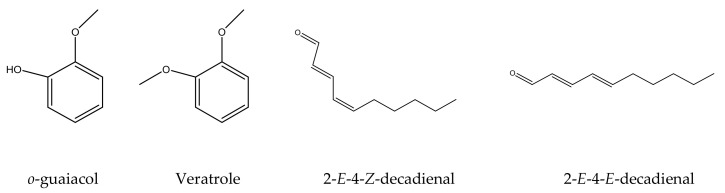
Chemical structures of the identified birch biomarkers.

**Table 1 plants-11-02132-t001:** Available information on commercial sweet birch samples.

Sample ID	Name	Sample Description	Botanical Name
C1	Birch	100% Pure Essential oil	*Betula* sp.
C2	Birch Sweet	100% Pure Essential oil	*Betula lenta*
C3	Birch	100% pure and natural Therapeutic Grade Essential oil	*Betula lenta*
C4	Birch	100% pure Essential oil	*Betula lenta*
C5	Birch	100% Pure Essential oil	*Betula lenta*
C6	Birch	100% pure Therapeutic Grade Essential oil	*Betula lenta*
C7	Birch	100% pure Therapeutic Grade Essential oil	*Betula lenta*
C8	Birch	Therapeutic Grade Essential oil	NA
C9	Birch	100% pure Therapeutic Grade Essential oil	*Betula lenta*
C10	Birch	100% pure Essential oil, Therapeutic Grade	*Betula lenta*
C11	Birch Sweet	100% pure Therapeutic Grade Essential oil	*Betula lenta*
C12	Birch	100% pure Essential oil	*Betula lenta*
C13	Birch Sweet	Pure Essential oil	*Betula lenta*
C14	Sweet Birch	100% pure Therapeutic Grade Essential oil	*Betula lenta*
C15	Birch Sweet	100% pure Therapeutic Grade Essential oil	*Betula lenta*
C16	Birch	100% Pure Essential oil	*Betula lenta*
C17	Birch premium	100% pure Therapeutic Grade Essential oil	*Betula lenta*
C18	Birch	100% Pure Essential oil	*Betula lenta*
C19	Birch	100% pure Essential oil	*Betula lenta*
C20	Birch	100% pure Essential oil	*Betula lenta*
C21	Birch	100% Pure Essential oil	*Betula* sp.
C22	Birch	100% Pure Essential oil	*Betula lenta*
C23	Birch	100% Pure Essential oil	*Betula lenta*
C24	Birch	100% Pure Essential oil	*Betula lenta*
C25	Birch	100% pure Essential oil	*Betula lenta*
C26	Birch	100% pure Essential oil	*Betula lenta*
C27	Birch	100% pure Essential oil	*Betula lenta*

**Table 2 plants-11-02132-t002:** Chemical composition of authentic (A) sweet birch EO samples.

RI_calc_	RI_exp_	Compounds	A1	A2	A3	A4	A5	A6	A7	A8	A9	A10	A11	A12	A13	A14	A15	A16	A17	A18	A19	A20	A21
801	802	Hexanal	-	-	0.01	0.01	0.01	-	-	-	-	-	-	0.01	0.01	0.02	tr	0.02	0.05	0.05	0.06	0.01	0.01
846	846	2*E*-Hexenal	-	-	-	-	-	-	-	-	-	-	-	-	-	0.13	0.02	-	-	-	-	-	-
862	862	n-Hexanol	-	-	-	-	-	-	-	-	-	-	-	-		0.01	-	-	0.01	0.02	0.01	-	-
934	931	α-Pinene	-	-	-	0.01	0.01	-	-	-	-	-	-	tr	-	-	-	-	0.01	-	-	-	-
970	971	Sabinene	-	-	-	-	-	-	-	-	-	-	-	-	-	0.01	-	-	-	-	-	-	-
975	974	β-Pinene	-	-	-	0.01	0.01	-	-	-	-	-	-	0.01	tr	tr	-	-	-	-	-	-	-
977	977	Phenol	-	-	-	-	-	-	-	-	-	0.01	0.01	-	-	-	-	-	-	-	-	-	-
988	984	2-Pentyl furan	-	0.01	-	0.01	0.02	tr	-	-	-	tr	-	0.01	0.02	-	-	0.04	0.01	0.01	-	-	-
1009	1009	*o*-Methyl anisole	-	-	-	0.01	0.01	-	-	-	-	-	-	tr	-	-	-	-	-	-	-	-	-
1024	1020	*p*-Cymene	-	-	-	-	--	-	-	-	-	--	-	-	-	tr	-	0.02	-	0.01	-	-	-
1044	1044	*E*-β-Ocimene	-	-	-	-	-	-	-	-	-	-	-	-	-	0.01	-	-	-	-	-	-	-
1050	1048	*o*-Cresol	-	-	-	-	-	-	-	tr	-	-	tr	-	-	tr	-	-	0.01	0.01	0.03	-	-
1068	1070	n-Octanol	-	-	-	-	-	-	-	tr	-	-	-	-	-	-	-	-	0.08	0.04	0.03	-	-
1087	1087	*o*-Guaiacol	tr	tr	tr	tr	tr	0.02	tr	0.04	0.04	0.06	0.10	0.01	tr	0.04	0.05	tr	0.47	0.30	0.39	0.02	0.02
1095	1098	Linalool	-	-	-	-	-	-	-	tr	-	-	-	-	-	0.01	tr	-	0.13	0.01	0.04	-	-
1100	1103	n-Nonanal	-	0.02	0.02	0.02	0.02	-	-	tr	-	-	-	0.01	0.01	0.02	0.01	-	-	0.01	-	0.01	0.01
1141	1142	Veratrole	tr	tr	tr	0.01	0.01	tr	tr	0.01	tr	0.01	0.02	0.01	tr	0.01	0.01	0.02	tr	tr	tr	tr	0.01
1163	1160	2-*E*-Nonenal	-	0.01	-	0.01	0.01	-	-	-	-	tr	-	-	tr	-	-	-	-	-	-	0.03	0.01
1177	1180	Terpinen-4-ol	-	-	-	-	-	-	-	-	-	-	-	-	-	0.01	-	0.02	0.02	-	0.01	-	-
1191	1191	Methyl salicylate	98.97	97.37	98.96	98.30	98.22	99.54	99.78	99.77	99.84	98.70	99.11	98.38	97.93	99.48	99.67	93.24	97.74	99.20	98.82	97.87	98.49
1195	1195	Methyl chavicol	-	-	-	-	-	-	-	-	-	-	-	-	-	-	-	-	0.12	-	-	-	-
1201	1201	n-Decanal	-	0.01	0.01	0.01	0.01	-	-	tr	-	-	-	0.01	0.01	tr	tr	-	-	-	-	0.01	-
1239	1236	*o*-Anisaldehyde	-	-	-	0.01	0.01	-	-	-	-	0.01	0.01	0.01	0.01	-	-	-	-	-	-	-	-
1260	1260	2-*E*-Decenal	-	-	-	-	tr	-	-	-	-	-	-	0.01	0.01	-	-	-	-	-	-	tr	-
1266	1266	Ethyl salicylate	0.58	0.76	0.49	0.61	0.61	0.06	0.05	0.04	0.03	0.63	0.33	0.60	0.59	0.03	0.03	6.41	0.74	0.08	0.40	1.7	0.84
1283	1282	Bornyl acetate	-	-	-	0.01	0.01	-	-	-	-	-	-	0.01	-	-	-	-	-	-	-	-	-
1292	1291	2-*E*-4-*Z*-Decadienal	0.04	0.10	0.10	0.05	0.06	0.02	tr	tr	tr	tr	tr	0.05	0.09	tr	tr	tr	tr	tr	tr	0.01	0.01
1319	1315	2-*E*-4-*E*-Decadienal	0.06	0.21	0.15	0.12	0.12	0.06	0.01	tr	tr	0.02	tr	0.11	0.19	0.01	tr	tr	tr	tr	tr	0.02	0.02
1322	1319	Methyl geranate	-	-	-	-	-	-	-	-	-	-	-	0	0.01	-	-	-	-	-	-	-	-
1332	1332	Methyl *o*-anisate	0.29	0.40	0.15	0.53	0.45	0.26	0.13	0.04	0.04	0.54	0.39	0.48	0.34	0.04	0.08	0.09	0.44	0.08	0.14	0.12	0.19
1356	1345	Eugenol	-	-	-	-	-	-	0.01	0.03	0.02	tr	0.01	-	-	0.03	0.02	-	0.01	0.04	0.02	tr	-
1365	1365	*Z*-8-Undecenal	-	0.01	0.02	-	-	-	-	-	-	-	-	-	0.01	-	-	-	-	-	-	-	-
1382	1382	Hexyl hexanoate	-	-	-	-	-	-	-	-	-	-	-	-	-	0.01	-	-	-	-	-	-	-
1417	1417	β-Caryophyllene	-	0.01	-	-	0.01	-	-	tr	-	-	-	tr	0	0.01	0.01	-	-	-	-	-	-
1423	1423	Isobutyl salicylate	-	0.04	-	0.01	0.02	-	-	-	-	-	-	0.01	0.04	-	-	-	-	-	-	0.03	0.01
1453	1445	Geranyl acetone	-	0.01	-	tr	0.01	-	-	-	-	-	-	0.01	0.01	-	-	-	-	-	-	-	0.01
1452	1453	α-Humulene	-	0.03	-	0.02	0.02	-	-	tr	-	-	-	0.02	0.02	tr	0.01	-	-	-	-	-	-
1432	1480	*E*-β-Bergamotene	-	-	-	-	-	-	-	-	-	-	-	-	tr	-	-	-	-	-	-	-	-
1500	1495	α-Muurolene	-	0.02	0.01	-	-	-	-	-	-	-	-	tr	0.01	-	-	-	-	-	-	-	-
1505	1502	*E-E*-α-Farnesene	-	-	-	-	-	-	-	-	-	-	-	-	-	0.03	-	-	-	-	-	-	-
	1507	Unidentified	-	-	-	-	0.02	0.01	-	-	-	-	-	0.02	-	-	0.09	-	-	-	-	-	-
1535	1528	Isopentyl salicylate	0.01	0.04	0.01	0.05	0.09	-	-	-	-	-	-	0.06	0.03	-	-	0.12	-	-	-	-	0.01
1568	1568	3-*E*-Hexenyl benzoate	-	-	-	-	-	-	-	-	-	-	-	-	-	0.02	0.01	-	-	-	-	-	-
1574	1574	Pentyl salicylate	-	-	-	-	-	-	-	-	-	-	-	0.01	0.01	-	-	-	-	-	-	-	-
1579	1575	n-Hexyl benzoate	-	-	-	-	-	-	-	-	-	-	-	-	-	0.01	-	-	-	-	-	-	-
1577	1577	Caryophyllene oxide	-	-	-	-	-	-	-	tr	-	-	-	-	-	0.01	0.01	-	-	-	-	-	-
1679	1679	Hexyl salicylate	-	-	-	-	-	-	-	-	-	-	-	-	0.01	-	-	-	-	-	-	-	-
1697	1696	2-Pentadecanone	-	-	-	-	-	-	-	-	-	-	-	-	tr	-	-	-	-	-	-	-	-
1700	1699	Heptadecane	-	-	-	-	0.01	-	-	-	-	-	-	tr	0.01	tr	-	-	-	-	-	-	-
1714	1714	Pentadecanal	-	0.03	0.01	0.01	0.01	0.02	tr	-	-	0.01	0.01	0.01	0.02	-	-	-	tr	-	-	-	-
1800	1799	Octadecane	-	0.01	tr	tr	0.01	-	-	-	-	-	-	0.01	0.01	tr	tr	-	-	0.01	-	tr	-
1864	1864	Benzyl salicylate	-	0.02	-	0.01	0.01	-	tr	0.01	tr	tr	-	0.01	0.03	-	-	-	-	-	-	0.01	0.01
1900	1899	Nonadecane	-	0.01	0.01	0.01	0.01	-	-	tr	-	-	-	0.01	0.01	tr	tr	-	-	0.01	-	0.01	-
1903	1903	*E-E*,-5,9,13-Pentadecatrien-2-one, 6,10,14-trimethyl	-	-	-	-	0.01	-	-	-	-	-	-	0	0.01	-	-	-	-	-	-	-	-
2000	2000	Eicosane	0.01	0.03	0.01	0.02	0.02	-	tr	tr	tr	-	-	0.02	0.02	-	tr	-	-	0.01	-	-	0.01
2061	2061	Phenyl ethyl alcohol dimer	-	-	-	-	-	-	-	tr	-	-	-	tr	-	tr	-	-	-	-	-	0.01	-
2100	2100	Heneicosane	0.01	0.03	0.01	0.02	0.02	-	0.01	0.01	0.01	tr	-	0.02	0.02	tr	0.01	-	-	0.02	-	0.01	tr
2138	2138	Oxacycloheptadecenone	-	-	-	-	-	-	-	-	-	-	-	0.01	0.01	-	-	-	-	-	-	-	-
2200	2199	Docosane	-	-	-	-	-	-	-	-	-	-	-	-	0.01	-	-	-	-	-	-	-	-
2300	2300	n-Tricosane	-	-	-	0.01	-	-	-	tr	-	-	-	-	-	-	0.01	-	-	-	-	-	-
2500	2498	Pentacosane	-	-	-	-	-	-	-	0.01	-	-	-	-	-	-	0.01	-	-	-	-	-	-
2700	2698	Heptacosane	-	-	-	-	-	-	-	-	-	-	-	-	-	-	0.01	-	-	-	-	-	-
		Yield (%)	0.07	0.1	0.08	0.07	0.09	0.09	0.07	0.07	0.07	0.1	0.07	0.07	0.08	0.1	0.08	0.1	0.1	0.08	0.07	0.07	0.1

**Table 3 plants-11-02132-t003:** Comparison of authentic wintergreen EOs and authentic birch EOs.

Compound Name	Wintergreen (34 Samples)			Birch (21 Samples)		
	Range (%)	Mean	SD	Range (%)	Mean	SD
1,8-Cineole	tr–0.05	0.01	0.01	-	-	-
2-Pentyl furan	-	-	-	0–0.04	0.02	0.01
2-*E*-4-*Z*-Decadienal	-	-	-	tr–0.10	0.08	0.03
2-*E*-4-*E*-Decadienal	-	-	-	tr–0.21	0.05	0.07
2-*E*-Decenal	-	-	-	0–0.01	0.01	0
2*E*-Hexenal	-	-	-	0–0.13	0.08	0.08
2-*E*-Nonenal	-	-	-	0–0.01	0.01	0.01
3-*Z*-Hexenol	0.01–0.03	0.01	0.01	-	-	-
3-*Z*-Hexenyl 3-methyl butanoate	tr	tr	0	-	-	-
3-*Z*-Hexenyl butanoate	0.01	0.01	0	-	-	-
3-Methyl-1,2-cyclopentanedione	tr–0.01	0.01	0	-	-	-
3-*E*-Hexenyl benzoate	-	-	-	0–0.02	0.02	0.01
*allo*-Ocimene	0.01	0.01	0	-	-	-
α-Humulene	tr	tr	0	0–0.03	0.02	0.01
α-Muurolene	-	-		0–0.02	0.02	0.01
α-Phellandrene	tr–0.01	tr	0.01	-	0.01	-
α-Pinene	tr–0.09	0.02	0.02	0–0.01	0.01	0
α-Thujene	tr	tr	0	-	-	-
Artemisia alcohol	tr	tr	0	-	-	-
Benzaldehyde	tr–0.01	0.01	0.01	-	-	-
Benzyl alcohol	tr–0.02	0.01	0.01	-	-	-
Benzyl salicylate	-	-	-	0–0.03	0.01	0.01
β-Caryophyllene	tr–0.02	0.01	0	0–0.01	0.01	0
β-Dehydro elsholtzia ketone	tr–0.01	0.01	0	-	-	-
β-Pinene	tr–0.05	0.02	0.01	0–0.01	0.01	0
Bornyl acetate	tr–0.01	0.01	0	0–0.01	0.01	0
Camphene	tr–0.01	0.01	0.01	-	-	-
Caryophyllene oxide	-	-	-	0–0.01	0.01	0
*Z*-8-Undecenal	-	-	-	0–0.02	0.01	0.01
δ-Cadinene	tr	tr	0	-	-	-
Docosane	-	-	-	0–0.01	0.01	0
Eicosane	-	-	-	0–0.03	0.02	0.01
Ethyl benzoate	tr–0.01	0.01	0	-	-	-
Ethyl salicylate	tr–0.33	0.1	0.07	0.03–6.41	0.74	1.36
Eugenol	0.01–0.14	0.05	0.03	0–0.04	0.03	0.01
Geraniol	tr–0.01	0.01	0	-	-	-
Geranyl acetone	-	-	-	0–0.01	tr	0
Germacrene D	tr	tr	0	-	-	-
Heneicosane	-	-	-	0 -0.03	0.02	0.01
Heptacosane	-	-	-	0–0.01	0.01	0
Heptadecane	-	-	-	0–0.01	0.01	0
Hexanal	-	-	-	0–0.06	0.02	0.02
Hexenyl acetate	0.01	0.01	0	-	-	-
Hexyl hexanoate	-	-	-	0–0.01	0.01	0
Hexyl salicylate	-	-	-	0–0.01	0.01	0
Hotrienol	tr	tr	0	-	-	-
Isobutyl salicylate	-	-	-	0–0.04	0.01	0.01
Isopentyl salicylate	-	-	-	0–0.09	0.05	0.04
Isothymol	tr–0.01	0.01	0	-	-	-
Limonene	tr–0.02	0.01	0.01	-	-	-
Linalool	0–0.06	0.03	0.01	0–0.13	0.05	0.06
Menthone	tr–0.02	0.02	0	-	-	-
Methyl chavicol	-	-	-	0–0.12	0.10	0
Methyl geranate	-	-	-	0–0.01	0.01	0.01
Methyl o-anisate	tr–0.01	0.01	0	0.04–0.54	0.25	0.18
Methyl salicylate	99.47–100	99.77	0.13	93.24–99.84	98.54	1.42
n-Decanal	-	-	-	0–0.01	0.01	0
n-Hexanol	tr–0.01	0.01	0	0–0.02	0.01	0
n-Hexyl benzoate	-	-	-	0–0.01	0.01	0
n-Nonanal	tr–0.01	tr	0.01	0–0.02	0.01	0
n-Octane	tr	tr	0	-	-	-
n-Octanol	tr–0.01	0.01	0.01	0–0.08	0.05	0.03
Nonadecane	-	-	-	0–0.01	0.01	0
n-Tricosane	-	-	-	0–0.01	0.01	0
Octadecane	-	-	-	0–0.01	0.01	0
o-Anisaldehyde	-	-	-	0–0.01	0.01	0
o-Cresol	-	-	-	0–0.03	0.01	0.01
o-Cymene	tr	tr	0	-	-	-
o-Guaiacol	-	-	-	tr–0.47	0.12	0.16
o-Methyl anisole	-	-	-	0–0.01	0.01	0
Oxacycloheptadecenone	-	-	-	0–0.01	0.01	0
p-Mentha-1(7),8(10)-dien-9-ol	tr–0.01	0.01	0	-	-	-
*p*-Xylene	tr	tr	0	-	-	-
*p*-Cymene	tr–0.01	0.01	0.01	0–0.02	0.02	0.01
Pentacosane	-	-	-	0–0.01	0.01	0
Pentadecanal	-	-	-	0–0.03	0.01	0.01
Pentyl salicylate	-	-	-	0–0.01	0.01	0
Phenol	tr–0.01	0.01	0	0–0.01	0.01	0
Phenyl ethyl alcohol dimer	tr	tr	0	-	-	-
Pulegone	tr–0.01	0.01	0	-	-	-
Sabinene	-	-	-	0–0.01	0.01	0
Terpinen-4-ol	tr	tr	0	0–0.02	0.02	0.01
Thymol	tr–0.01	0.01	0	-	-	-
Toluene	tr–0.01	0.01	0.01	-	-	-
E-β-Ocimene	-	-	-	0–0.01		0
E-Caryophyllene	tr–0.01	0.01	0	-	-	-
E-E,-5,9,13-Pentadecatrien-2-one, 6,10,14-trimethyl	-	-	-	0–0.01	0.01	0
E-E-α-Farnesene	-	-	-	0–0.03	0.02	0
Veratrole	-	-	-	tr–0.02	0.01	0
Vitispirane	tr–0.02	0.01	0.01	-	-	-
Wintergreen sesquiterpenoid 1	tr–0.01	0.01	0	-	-	-
Xylene isomer	tr	tr	0	-	-	-

**Table 4 plants-11-02132-t004:** Chemical composition of commercial (C) birch EO samples.

RI_calc_	RI_exp_	Compounds	C1	C2	C3	C4	C5	C6	C7	C8	C9	C10	C11	C12	C13	C14	C15	C16	C17	C18	C19	C20	C21	C22	C23	C24	C25	C26	C27
848	848	3-*Z*-Hexenol	-	-	0.02	0.03	-	-	-	-	0.05	-	-	-	-	-	-	-	-	-	-	-	-	-	-	-	-	-	-
862	862	n-Hexanol	-	-	tr	0.01	-	-	-	-	0.01	-	-	-	-	-	-	-	-	-	-	-	-	-	-	-	-	-	-
895	895	o-Xylene	-	-	-	-	-	-	-	-	-	-	-	-	-	1.36	-	-	-	-	-	-	-	-	-	-	-	-	-
900	900	n-Nonane	-	-	-	-	-	-	-	-	-	-	0.01	-	-	0.04	-	-	-	-	-	-	-	-	-	-	-	-	-
920	919	Hashishene	-	-	-	-	-	-	-	-	-	0.01	-	-	-	-	-	-	-	-	-	-	-	-	-	-	-	-	-
	921	Tricyclene + Hexylene glycol	-	-	-	-	-	-	-	-	-	-	-	-	-	0.14	-	-	-	-	-	-	-	-	-	-	-	-	-
921	921	Tricyclene	-	-	-	-	-	-	-	0.02	-	-	-	-	-	-	-	-	-	-	-	-	-	-	-	0.04	-	-	-
924	924	α-Thujene	-	-	-	tr	-	-	-	-	-	2.93	-	-	0.01	0.06	-	-	-	-	-	-	-	-	-	-	0.27	-	-
934	931	α-Pinene	-	-	0.01	0.04	-	-	0.01	0.28	0.02	0.21	-	-	0.19	6.11	-	0.18	2.30	-	-	-	-	0.01		1.86	0.93	0.01	-
942	942	Thujadiene	-	-	-	-	-	-	-	-	-	0.02	-	-	-	-	-	-	-	-	-	-	-	-	-	-	-	-	-
993	945	α-Fenchene	-	-	-	-	-	-	-	0.04	-	-	-	-	-	0.05	-	-	-	-	-	-	-	-	-	-	-	-	-
949	948	Camphene	-	-	-	-	-	-	-	0.14	-	-	-	-	0.02	0.36	-	0.03	0.50	-	-	-	-	-	-	0.65	-	-	-
969	970	3,7,7-Trimethyl-1,3,5-cycloheptatriene	-	-	-	-	-	-	-	0.01	-	-	-	-	-	0.02	-	-	-	-	-	-	-	-	-	-	-	-	-
970	971	Sabinene	-	-	-	-	-	-	-	-	-	0.18	-	-	0.01	0.10	-	-	-	-	-	-	-	-	-	1.31	-	-	-
975	974	β-Pinene	-	-	0.01	0.02	-	-	tr	0.03	-	0.01	-	-	0.07	0.17	-	0.02	2.28	0.03	-	-	-	tr	-	-	-	-	-
988	988	Myrcene	-	-	-	tr	-	-	tr	0.06	-	0.02	-	-	0.02	1.66	-	tr	-	-	-	-	-	-	-	-	-	-	-
1001	999	δ-2-Carene	-	-	-	-	-	-	-	0.01	-	-	-	-	-	0.19	-	-	-	-	-	-	-	-	-	-	-	-	-
1006	1006	α-Phellandrene	-	-	-	-	-	-	-	0.01	-	0.04	-	-	tr	1.94	-	-	-	-	-	-	-	-	-	-	-	-	-
1008	1008	δ-3-Carene	-	-	-	0.01	-	-	-	1.92	-	0.13	-	-	-	8.42	-	-	0.24	0.04	-	-	-	-	-	1.36	-	-	-
1012	1012	1,4-Cineole	-	-	-	-	-	-	-	0.14	-	-	-	-	-	0.30	-	-	-	-	-	-	-	-	-	-	-	-	-
1014	1014	α-Terpinene	-	-	-	-	-	-	-	0.04	-	0.01	-	-	0.09	2.57	-	-	-	-	-	-	-	-	-	-	-	-	-
1020	1020	*p*-Cymene	-	-	0.01	0.01	-	-	-	0.05	-	0.05	-	-	0.12	0.68	-	0.01	-	-	-	-	-	-	-	-	-	-	-
1026	1026	Acetyl methyl furan	-	-	-	-	-	-	-	-	-	0.01	-	-	-	-	-	-	-	-	-	-	-	-	-	-	-	-	-
1024	1027	Limonene	-	-	0.01	0.01	0.05	0.06	0.02	0.28	0.01	0.06	-	-	0.26	1.39	-	0.04	0.55	0.09	0.02	-	-	0.01	-	0.47	0.27	0.01	-
1025	1029	β-Phellandrene	-	-	-	0.01	-	-	tr	-	-	0.02	-	-	-	0.19	-	-	-	-	-	-	-	-	-	0.15	-	-	-
1026	1031	1,8-Cineole	-	-	0.01	0.02	-	-	0.01	0.11	0.01	-	-	-	0.69	2.30	-	0.33	-	-	-	0.06	-	0.02	-	-	-	0.02	0.06
1032	1033	*Z*-β-Ocimene	-	-	-	-	-	-	-	-	-	0.01	-	-	-	0.12	-	-	-	-	-	-	-	-	-	-	-	-	-
	1041	Unidentified	-	-	-	-	-	-	-	-	-	-	-	-	-	0.48	-	-	-	-	-	-	-	-	-	-	-	-	-
1044	1044	E-β-Ocimene	-	-	-	-	-	-	-	-	-	-	-	-	-	0.24	-	-	-	-	-	-	-	-	-	-	-	-	-
	1055	Unidentified	-	-	-	-	-	-	-	-	-	-	-	-	-	0.25	-	-	-	-	-	-	-	-	-	-	-	-	-
1054	1058	γ-Terpinene	-	-	-	-	-	-	-	0.02	-	0.02	-	-	0.23	0.27	-	0.01	-	-	-	-	-	-	-	-	-	-	-
1063	1070	n-Octanol	-	-	-	-	0.01	0.01	-	-	-	-	-	-	-	-	-	-	-	-	-	-	-	-	-	-	-	-	-
1086	1088	Terpinolene	-	-	-	-	-	-	-	0.11	-	0.01	-	-	0.04	1.67	-	-	-	-	-	-	-	-	-	-	-	-	-
1086	1092	Fenchone	-	-	-	-	-	-	tr	-	-	-	-	-	-	0.04	-	-	-	-	-	-	-	tr	-	-	-	-	-
1095	1098	Linalool	-	0.03	0.03	0.04	0.01	0.02	0.03	-	0.05	-	-	0.02	0.08	-	-	0.03	-	0.04	0.03	0.07	0.04	0.03	0.03	-	-	0.03	0.06
1106	1106	Isocamphone	-	-	-	-	-	-	-	-	-	-	-	-	-	0.33	-	-	-	-	-	-	-	-	-	-	-	-	-
1106	1114	Z-Rose oxide	-	-	-	-	-	-	-	-	-	-	-	-	0.01	-	-	-	-	-	-	-	-	-	-	-	-	-	-
1112	1119	E-Thujone	-	-	-	-	-	-	0.01	-	-	0.01	-	-	-	-	-	-	-	-	-	-	-	0.01	-	-	-	-	-
1118	1121	Isophorone	-	-	-	-	-	-	-	-	-	-	-	-	-	2.96	-	-	-	-	-	-	-	-	-	-	-	-	-
1130	1131	Terpin-3-en-1-ol	-	-	-	-	-	-	-	-	-	-	-	-	-	0.22	-	-	-	-	-	-	-	-	-	-	-	-	-
1141	1145	Camphor	-	tr	-	-	-	-	-		-	-	-	-	0.03	4.54	-	0.09	0.54	-	-	-	-	-	-	-	-	-	-
1143	1150	Z-β-Terpineol	-	-	-	-	-	-	-	-	-	-	-	-	-	0.13	-	-	-	-	-	-	-	-	-	-	-	-	-
1148	1152	Citronellal	-	-	-	-	-	-	-	-	-	-	-	-	0.01	-	-	-	-	-	-	-	-	-	-	-	-	-	-
1148	1157	Menthone	-	-	-	-	-	-	-	-	-	0.01	-	-	0.01	0.04	-	-	-	-	-	-	-	-	-	-	-	-	-
1155	1162	Isoborneol	-	-	-	-	-	-	-	-	-	-	-	-	-	0.2	-	-	-	-	-	-	-	-	-	-	-	-	-
1157	1163	Benzyl acetate	-	0.02	-	-	-	-	-	-	-	-	-	0.01	-	-	-	-	-	-	-	-	-	-	-	-	-	-	-
1166	1166	iso-Menthone	-	-	-	-	-	-	-	-	-	-	-	-	0.05	-	-	-	-	-	-	-	-	-	-	-	-	-	-
1169	1170	Borneol	-	-	-	-	-	-	-	-	-	-	-	-	-	0.14	-	0.02	-	-	-	-	-	-	-	-	-	-	-
1176	1175	Menthol	-	-	-	-	-	-	-	-	-	0.01	-	-	-	-	-	-	-	-	-	-	-	-	-	-	-	-	-
1177	1180	Terpinen-4-ol	-	-	-	-	0.01	0.01	-	-	-	0.01	-	-	0.47	0.12	-	0.01	-	-	-	-	-	-	-	-	-	-	-
1191	1191	Methyl salicylate	99.99	99.90	99.80	99.64	99.38	99.29	99.78	95.18	99.34	95.64	99.68	99.91	96.67	56.71	99.95	98.89	93.37	99.68	99.95	99.82	99.87	99.85	99.96	93.83	97.58	99.74	99.78
1195	1195	Methyl chavicol	-	-	-	-	-	-	-	-	-	0.21	-	-	-	-	-	-	-	-	-	-	-	-	-	-	-	-	-
1196	1196	Isocamphenone	-	-	-	-	-	-	-	-	-	-	-	-	-	0.26	-	-	-	-	-	-	-	-	-	-	-	-	-
1199	1197	γ-Terpineol	-	-	-	-	-	-	-	-	-	-	-	-	-	0.35	-	-	-	-	-	-	-	-	-	-	-	-	-
1209	1209	Octyl acetate	-	-	-	-	0.01	0.02	-	-	-	-	-	-	-	-	-	-	-	-	-	-	-	-	-	-	-	-	-
1249	1250	Geraniol	-	-	-	-	tr	0.01	0.01	-	-	-	-	-	-	-	-	-	-	-	-	-	-	-	-	-	-	0.03	-
1252	1252	Linalyl acetate	-	-	-	0.01	-	-	-	-	-	-	-	-	0.20	-	-	-	-	-	-	-	-	-	-	-	-	-	-
1266	1266	Ethyl salicylate	0.01	0.04	0.05	0.08	0.50	0.55	0.07	-	0.07	-	0.26	0.03	-	-	-	0.15	-	0.05	-	0.04	0.04	0.04	-	-	-	0.11	0.08
1271	1271	Citronellyl formate	-	-	-	-	-	-	-	-	-	-	-	-	0.08	-	-	-	-	-	-	-	-	-	-	-	-	-	-
1281	1281	Vitispirane	-	0.01	0.02	0.01	0.02	0.02	0.01	-	0.03	-	-	0.01	-	-	-	0.01	-	-	-	-	-	0.01	-	-	-	-	-
1283	1282	Bornyl acetate	-	-	-	-	-	-	-	-	-	-	-	-	0.02	0.65	-	0.08	0.20	0.07	-	-	-	Tr	-	0.15	-	-	-
1293	1293	Methyl naphthalene	-	-	-	0.01	-	-	-	-	-	-	-	-	-	-	-	-	-	-	-	-	-	-	-	-	-	-	-
1298	1297	Geranyl formate	-	-	-	-	-	-	-	-	-	-	-	-	0.02	-	-	-	-	-	-	-	-	-	-	-	-	-	-
1299	1299	β-Dehydro elsholtzia ketone	-	-	-	0.01	-	-	-	-	0.01	-	-	-	-	-	-	-	-	-	-	-	-	-	-	-	-	0.01	-
1346	1344	α-Terpinyl acetate	-	-	-	-	-	-	-	-	-	-	-	-	0.02	-	-	0.01	-	-	-	-	-	-	-	-	-	-	-
1356	1345	Eugenol	-	-	0.01	0.03	-	-	0.01	-	0.01	-	-	-	-	-	-	-	-	-	-	-	-	-	-	-	-	-	-
1371	1371	Longicyclene	-	-	-	-	-	-	-	-	-	-	-	-	-	0.04	-	-	-	-	-	-	-	-	-	-	-	-	-
1379	1376	Geranyl acetate	-	-	-	-	-	-	-	-	-	-	-	-	0.01	-	-	-	-	-	-	-	-	-	-	-	-	-	-
1387	1384	β-Bourbonene	-	-	-	-	-	-	-	-	-	0.01	-	-	0.01	-	-	-	-	-	-	-	-	-	-	-	-	-	-
1407	1407	Longifolene	-	-	-	-	-	-	-	-	-	-	-	-	-	0.15	-	-	-	-	-	-	-	-	-	-	-	-	-
1417	1417	β-Caryophyllene	-	0.01	tr	0.01	0.01	0.01	-	0.02	0.01	-	-	0.01	0.05	-	-	0.01	-	-	-	-	-	-	-	-	-	0.01	-
1439	1434	Aromadendrene	-	-	-	-	-	-	-	-	-	-	-	-	0.01	-	-	-	-	-	-	-	-	-	-	-	-	0.03	-
1452	1453	α-Humulene	-	-	-	-	-	-	-	0.17	-	-	-	-	0.0	-	-	-	-	-	-	-	-	-	-	-	-	-	-
1458	1457	allo-Aromadendrene	-	-	-	-	-	-	0.01	-	-	-	-	-	0.01	-	-	-	-	-	-	-	-	-	-	-	-	0.01	-
1461	1459	Z-Cadina-1(6),4-diene	-	-	-	-	-	-	-	0.01	-	-	-	-	-	-	-	-	-	-	-	-	-	-	-	-	-	-	-
1475	1470	E-Cadina-1(6),4-diene	-	-	-	-	-	-	-	0.02	-	-	-	-	-	-	-	-	-	-	-	-	-	-	-	-	-	-	-
1483	1476	α-Amorphene	-	-	-	-	-	-	-	0.01	-	-	-	-	-	-	-	-	-	-	-	-	-	-	-	-	-	-	-
1489	1487	β-Selinene	-	-	-	-	-	-	-	0.04	-	-	-	-	-	-	-	-	-	-	-	-	-	-	-	-	-	-	-
1496	1490	Viridiflorene	-	-	-	-	-	-	-	-	-	-	-	-	0.03	-	-	-	-	-	-	-	-	-	-	-	-	-	-
1489	1492	α-Selinene	-	-	-	-	-	-	-	0.06	-	-	-	-	-	-	-	-	-	-	-	-	-	-	-	-	-	-	-
1500	1493	Bicyclogermacrene	-	-	-	-	-	-	-	-	-	-	-	-	0.01	-	-	-	-	-	-	-	-	-	-	-	-	-	-
1500	1495	α-Muurolene	-	-	-	-	-	-	-	0.05	-	-	-	-	-	-	-	-	-	-	-	-	-	-	-	-	-	-	-
1509	1501	α-Bulnesene	-	-	-	-	-	-	-	0.02	-	-	-	-	-	-	-	-	-	-	-	-	-	-	-	-	-	-	-
	1514	Unidentified	-	-	-	-	-	-	-	0.04	-	-	-	-	-	-	-	-	-	-	-	-	-	-	-	-	-	-	-
1522	1517	δ-Cadinene	-	-	-	-	-	-	-	0.2	-	-	-	-	-	-	-	-	-	-	-	-	-	-	-	-	-	-	-
1521	1518	E-Calamenene	-	-	-	-	-	-	-	0.11	-	-	-	-	0.01	-	-	-	-	-	-	-	-	-	-	-	-	-	-
	1523	Unidentified	-	-	-	-	-	-	-	0.03	-	-	-	-	-	-	-	-	-	-	-	-	-	-	-	-	-	-	-
1533	1531	E-Cadine-1,4-diene	-	-	-	-	-	-	-	0.02	-	-	-	-	-	-	-	-	-	-	-	-	-	-	-	-	-	-	-
1544	1539	α-Calacorene	-	-	-	-	-	-	-	0.01	-	-	-	-	-	-	-	-	-	-	-	-	-	-	-	-	-	-	-
1548	1548	Isocaryophyllene oxide	-	-	-	-	-	-	-	0.08	-	-	-	-	-	-	-	-	-	-	-	-	-	-	-	-	-	-	-
1570	1570	Caryophyllenyl alcohol	-	-	-	-	-	-	-	0.02	-	-	-	-	-	-	-	-	-	-	-	-	-	-	-	-	-	-	-
1577	1577	Caryophyllene oxide	-	-	-	-	-	-	-	0.12	-	-	-	-	-	-	-	-	-	-	-	-	-	-	-	-	-	-	-
1610	1610	Dimethyl hydroxy terephthalate	-	-	-	-	-	-	-	0.33	0.01	0.35	-	-	0.39	0.09	0.05	-	-	-	-	-	-	-	-	-	-	-	-
2057	2057	Ricenalidic acid lactone	-	-	-	-	-	-	-	0.14	-	-	-	-	-	-	-	-	-	-	-	-	-	-	-	-	-	-	-
1282	1290	E-Anethole	-	-	-	-	-	-	-	-	-	-	-	-	-	-	-	-	-	-	-	-	0.03	-	-	-	-	-	-
1298	1300	Carvacrol	-	-	-	-	-	-	-	-	-	-	-	-	-	-	-	-	-	-	-	-	0.02	-	-	-	-	-	-
905	905	Cyclofenchene	-	-	-	-	-	-	-	-	-	-	-	-	-	-	-	-	-	-	-	-	-	-	0.01	-	-	-	-
884	907	Santene	-	-	-	-	-	-	-	-	-	-	-	-	-	-	-	-	-	-	-	-	-	-	-	0.18	-	-	-
1135	1123	Neral	-	-	-	-	-	-	0.01	-	-	-	-	-	-	-	-	-	-	-	-	-	-	-	-	-	-	-	0.02
1800	1800	Octadecane	-	-	-	-	-	-	-	-	-	-	0.01	-	-	-	-	-	-	-	-	-	-	-	-	-	-	-	-

## Data Availability

Not applicable.

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
