# Peer review of "Authentication and Market Survey of Sweet Birch (Betula lenta L.) Essential Oil"

_plants, 2022, doi:10.3390/plants11162132_

Round 1

Reviewer 1 Report

Comments on Manuscript ID : plants-1852462

 It is certain that the authenticity of the essential oils of Betula lenta offered on the market must be considered with the same attention as for those of wintergreen (Gaultheria spp.). Unlike the former, the authentication of the latter has been the subject of several works since those of Frey from Pepsico in 1986.[1]

The essential oils from the two species have a very similar composition, consisting of ca. 99% methyl salicylate. To authenticate the naturalness of this ester, it is easy to detect by GC-MS, preferably in SIM mode, the impurities that accompany the synthetic products acting as markers of the synthetic origins. [1][2][3][5]. The efficiency of this task is enhanced by the use of 13C, and 2H isotopic ratio measurements,[2] [5] and  quantitative  2H-ERETIC -NMR. [4]. Ultimately, 14C dating helps to ensure the contemporary origin of methyl salicylate [5]

Unfortunately, the authors have overlooked the article by Frey [1], and, most importantly,  the one by Murphy et al . [2].  They had also ignored the latter in their previous article on wintergreen oil,  published in this Journal earlier this year.[3] This is hardly acceptable since the article by Murphy et al. had been published (online) on 24/05/2021,  well  before the submission of the  manuscript on wintergreen to publication in Plants (17/03/2022).

 Whatsoever, it follows that the authentication techniques for 100% naturalness assessment in use for the essences of Gaultheria are equally applicable to those of Betula.

Regardless of these powerful yet sophisticated techniques, the authors consider that mere observation of the presence (or absence) of specific tracers in Betula essential oil would be necessary and sufficient for authenticity assessment. The reader can then understand that this determination also makes it possible to detect a mixture of the natural essential oil of Betula with that of wintergreen (lines 127-134), which is not possible with the 2H-ERETIC-NMR technique. Regrettably, the manuscript lacks information on this subject.

 The relative weakness of this second article submitted for publication in Plants may be  compounded by the anticipated counterproductive effect of this publication. Thanks to it, potential fraudsters will be informed about the tracers., which are easily available on the market. They may use them at low ppm levels to mimic the GC profile of the natural oil from Betula lenta.

 Miscellaneous remarks 

1. The information in the sentence (lines 34-37) in relation to reference [7] and in the sentence (lines 43-44) in relation to reference [9] is confusing and contradictory because the IUCN Red List of Threatened Species indicates that Betula lenta is listed as “least concern”

 2.  lines 13 and 105 : The apparent precision by the use of two decimal digits is illusory and inappropriate under the conditions of analysis by GC/FID with internal normalization. 

 3. line 53 : is wintergreen essential oil “a cheaper oil” ?; lines 52-54 : to be clarified.  4. line 57 : wrong statement. See reference [2]

 5. line 69 : The operating pressure of 6 bar (ca. 160°C) seems extravagant. Is it correct?

 6. Line 105-106: references [10] and [11] correspond to textbook knowledge and could be replaced by the citation of, for example, E. Guenther’s 6-volume book  (Vol VI. pages 3 and 11) 7. Since methyl ortho-anisate is present in all genuine samples (Table 2), it should be part of the list of  markers/tracers.

 8. Table 2

- o-guaiacol and 2-methoxyphenol correspond to the same chemical

- “(Z)-7-methyl-1,6-dioxa[4.5]decane” is certainly wrongly identified (no biogenetic likelihood) , and “(Z)” configuration corresponds to a double bond which does not exist.

- the likelihood of methyl p-mercaptobenzoate and “3-butyl-1,2,4-cyclopentanetrione” is mostly uncertain

- “phenyléthylalcohol dimer” and “oxacycloheptadecanone “ are undefined molecules. In the case of the latter, if it relates to  2-oxacycloheptadecan-1-one, it corresponds to  hexadecanolide.

________________________________________

[1] C. Frey,  Detection of synthetic flavorant addition to some essential oils by selected ion monitoring GC/MS, in Flavors and Fragrances: A World Perspective, B.M. Lawrence, B.D. Mookherjee and B.J. Willis (Editors),. Proceedings of the 10th International Congress of Essential Oils, Fragrances and Flavors, Washington, DC, U.S.A., 16-20 November 1986 © 1988 Elsevier Science Publishers B.V., Amsterdam; pp. 511-524

 [2] B. J. Murphy et al. Determining the authenticity of methyl salicylate in Gaultheria procumbens L. and Betula lenta L. essential oils using isotope ratio mass Spectrometry. J. Essent. Oil Res., 2021, 33 (5), 442-451.

 [3] P. K. Ojha et al, Volatile Constituent Analysis of Wintergreen Essential Oil and Comparison with Synthetic Methyl Salicylate for Authentication, Plants, 2022, 11,

[4] F. Le Grand et al. , J. Agric. Food Chem, 2005, 53, 5125-5129

 [5] A. Cuchet et al. Authentication of the naturalness of wintergreen (Gaultheria genus) essential oils by gas chromatography, isotope ratio mass spectrometry and radiocarbon assessment,, Ind. Crop. Prod., 2019, 142, 111873

Author Response

Comment 1: It is certain that the authenticity of the essential oils of Betula lenta offered on the market must be considered with the same attention as for those of wintergreen (Gaultheria spp.). Unlike the former, the authentication of the latter has been the subject of several works since those of Frey from Pepsico in 1986.[1] 

The essential oils from the two species have a very similar composition, consisting of ca. 99% methyl salicylate. To authenticate the naturalness of this ester, it is easy to detect by GC-MS, preferably in SIM mode, the impurities that accompany the synthetic products acting as markers of the synthetic origins. [1][2][3][5]. The efficiency of this task is enhanced by the use of 13C, and 2H isotopic ratio measurements,[2] [5] and quantitative 2H-ERETIC -NMR. [4]. Ultimately, 14C dating helps to ensure the contemporary origin of methyl salicylate [5] 

Unfortunately, the authors have overlooked the article by Frey [1], and, most importantly,  the one by Murphy et al . [2].  They had also ignored the latter in their previous article on wintergreen oil,  published in this Journal earlier this year.[3] This is hardly acceptable since the article by Murphy et al. had been published (online) on 24/05/2021,  well before the submission of the  manuscript on wintergreen to publication in Plants (17/03/2022). 

Response 1: We appreciate the reviewer’s encouraging, critical and constructive comments on this manuscript. We have taken them fully into account in the revision. The suggested references were added to the text. 

Comment 2:Whatsoever, it follows that the authentication techniques for 100% naturalness assessment in use for the essences of Gaultheria are equally applicable to those of Betula. Regardless of these powerful yet sophisticated techniques, the authors consider that mere observation of the presence (or absence) of specific tracers in Betula essential oil would be necessary and sufficient for authenticity assessment. The reader can then understand that this determination also makes it possible to detect a mixture of the natural essential oil of Betula with that of wintergreen (lines 127-134), which is not possible with the 2H-ERETIC-NMR technique. Regrettably, the manuscript lacks information on this subject. 

Response 2: Thank you for the suggestion. The information was added to the text as “The biomarkers are selected based upon three main criteria: (1) the marker should be commercially unavailable or too expensive which renders the adulteration process very costly, (2) The marker should be detected consistently in all the tested authentic EO samples, and (3) A birch EO marker should be found exclusively in birch EO, not in wintergreen and vice versa.” 

Comment 3:The relative weakness of this second article submitted for publication in Plants may be  compounded by the anticipated counterproductive effect of this publication. Thanks to it, potential fraudsters will be informed about the tracers., which are easily available on the market. They may use them at low ppm levels to mimic the GC profile of the natural oil from Betula lenta. 

Response 3: We appreciate the reviewer’s comment. We understand the concern, but it might have already been happening. This article has synthetic markers of synthetic methyl salicylate, wintergreen biomarkers, and birch EO biomarkers. By combining all those pieces of information, potential fraudsters can be easily detected. 

Comment 4: 1. The information in the sentence (lines 34-37) in relation to reference [7] and in the sentence (lines 43-44) in relation to reference [9] is confusing and contradictory because the IUCN Red List of Threatened Species indicates that Betula lenta is listed as “least concern” 

Response 4:Thank you for the comment. We’re aware of the least concern status of B. lenta. Our manuscript does NOT claim any endangered or threatened status for B. lenta. We simply cited the IUCN for the plant’s distribution in USA.  

Comment 5:  2.  lines 13 and 105 : The apparent precision by the use of two decimal digits is illusory and inappropriate under the conditions of analysis by GC/FID with internal normalization.  

Response 5:Thank you for the comment. We disagree.  

Comment 6: 3. line 53 : is wintergreen essential oil “a cheaper oil” ?; lines 52-54 : to be clarified. 4. line 57 : wrong statement. See reference [2] 

Response 6:Thank you for the comment. Yes, wintergreen EO is a cheaper oil compared to true birch EO. Reference 2 does not mention any studies performed on the EO composition of B. lenta. 

Comment 7: 5. line 69 : The operating pressure of 6 bar (ca. 160°C) seems extravagant. Is it correct? 

Response 7:Yes, it is correct. 

Comment 8:6. Line 105-106: references [10] and [11] correspond to textbook knowledge and could be replaced by the citation of, for example, E. Guenther’s 6-volume book  (Vol VI. pages 3 and 11)7. Since methyl ortho-anisate is present in all genuine samples (Table 2), it should be part of the list of  markers/tracers. 

Response 8:Thank you for the comment. Ethyl salicylate and methyl ortho-anisate are common markers in both oils; however, they are present in higher amounts in sweet birch EO. 

Comment 9:8. Table 2 

- o-guaiacol and 2-methoxyphenol correspond to the same chemical 

- “(Z)-7-methyl-1,6-dioxa[4.5]decane” is certainly wrongly identified (no biogenetic likelihood) , and “(Z)” configuration corresponds to a double bond which does not exist.  

- the likelihood of methyl p-mercaptobenzoate and “3-butyl-1,2,4-cyclopentanetrione” is mostly uncertain 

- “phenyléthylalcohol dimer” and “oxacycloheptadecanone “ are undefined molecules. In the case of the latter, if it relates to  2-oxacycloheptadecan-1-one, it corresponds to  hexadecanolide. 

Response 9:Thank you for the observation. O-guaiacol was corrected in the text. Please refer to https://pubchem.ncbi.nlm.nih.gov/compound/12571167 and https://www.researchgate.net/publication/229863934_E-7-Methyl-16-dioxaspiro45decane_in_the_chemical_communication_of_European_Scolytidae_and_Nitidulidae_Coleoptera for (Z)-7-methyl-1,6-dioxaspiro[4.5]decane. Our library has identified these molecules with a confidence level > 90%. 

________________________________________ 

[1] C. Frey,  Detection of synthetic flavorant addition to some essential oils by selected ion monitoring GC/MS, in Flavors and Fragrances:A World Perspective, B.M. Lawrence, B.D. Mookherjee and B.J. Willis (Editors),. Proceedings of the 10th International Congress of Essential Oils, Fragrances and Flavors, Washington, DC, U.S.A., 16-20 November 1986 © 1988 Elsevier Science Publishers B.V., Amsterdam; pp. 511-524 

 [2] B. J. Murphy et al. Determining the authenticity of methyl salicylate in Gaultheria procumbens L. and Betula lenta L. essential oils using isotope ratio mass Spectrometry. J. Essent. Oil Res., 2021, 33 (5), 442-451. 

 [3] P. K. Ojha et al, Volatile Constituent Analysis of Wintergreen Essential Oil and Comparison with Synthetic Methyl Salicylate for Authentication, Plants, 2022, 11, 

[4] F. Le Grand et al. , J. Agric. Food Chem, 2005, 53, 5125-5129 

 [5] A. Cuchet et al. Authentication of the naturalness of wintergreen (Gaultheria genus) essential oils by gas chromatography, isotope ratio mass spectrometry and radiocarbon assessment,, Ind. Crop. Prod., 2019, 142, 111873 

The suggested references were added to the text. 

Reviewer 2 Report

Dear Authors,

The manuscript ID: plants-1852462_v1 entitled Authentication and market survey of sweet birch (Betula lenta L.) essential oil” written by Noura S. Dosoky, Ambika Poudel, and Prabodh Satyal is interesting.

The whole manuscript (Introduction, Materials and Methods, Results and Discussion, Conclusions) is properly organized. The introduction is concise and provides general data on Betula species, especially B. lenta L., its composition and application. The purpose of the work is concrete – assessment of the composition of the EO extracted from the bark of B. lenta obtained from trusted sources and comparing the composition with oils available in the US market. The results are presented in the form of tables and interpreted. Based on them, a brief discussion and conclusions were drawn.

In my opinion, the article is valuable, but unfortunately there are no additional studies to evaluate the biological activity of B. lenta L. essential oils. According to me, the results are not sufficient to be accepted and published in such a prestigious journal as „Plants”.

With highest regards,

Author Response

Comment 1: 

Dear Authors, The manuscript ID: plants-1852462_v1 entitled „Authentication and market survey of sweet birch (Betula lenta L.) essential oil” written by Noura S. Dosoky, Ambika Poudel, and Prabodh Satyal is interesting. The whole manuscript (Introduction, Materials and Methods, Results and Discussion, Conclusions) is properly organized. The introduction is concise and provides general data on Betula species, especially B. lenta L., its composition and application. The purpose of the work is concrete – assessment of the composition of the EO extracted from the bark of B. lenta obtained from trusted sources and comparing the composition with oils available in the US market. The results are presented in the form of tables and interpreted. Based on them, a brief discussion and conclusions were drawn. In my opinion, the article is valuable, but unfortunately there are no additional studies to evaluate the biological activity of B. lenta L. essential oils. According to me, the results are not sufficient to be accepted and published in such a prestigious journal as „Plants”. 

With highest regards, 

Response 1:We appreciate the reviewer’s critical and constructive comments on this manuscript. We have taken them fully into account in the revision. The biological activity of B. lenta L. essential oil is part of our future work. 

Reviewer 3 Report

Dear Authors,

The article review: „ Authentication and market survey of sweet birch (Betula lenta L.) essential oil”. The article presented the the EO extracted from the bark of B. lenta obtained from trusted sources and compares the composition to the oils available in the US market. The EOs of B. lenta were examined to identify chemical markers to help with authentication or adulteration detection. Authors identified: ortho-guaiacol, veratrole, 2-trans-4-cis-decadienal, and 2-trans-4-trans-decadi-enal as natural marker compounds for authentic B. lenta oil. The set goals of the research have been achieved. The presented research results are very interesting and worth publishing after correction of the manuscript.

Main comments:

·         Title „ Authentication and market survey of sweet birch (Betula lenta L.) essential oil”is adequate to the content.

·         The purpose of the research should be better specified.

·         The same name of the particular EOs components should be use in the text and in the tables; different names are currently provided, for example ortho-guaiacol/o-Guaiacol.

·         Table 4 should be corrected - values ​​take 2 rows.

·         In lines 62 and 63 state: “Sweet birch bark essential oils from authentic sources (hydro-distilled in both lab and industrial settings)”. It is not clear; which samples were hydro-distilled in laboratory and which in industrial settings. These are definitely different conditions and a different results. Should be better explained in section Materials and Methods.

·         SI units should be given.

·         The sites of the natural samples of B. lenta should be better characterized. Generally, it is not known where the plants/bark came from.

·         EO yield should be presented for each samples.

·         Literature should be cited after journal requirements (italics of latin names of plant species, unnecessary spaces, etc.).

Author Response

Reviewer 3 

Comment 1: Dear Authors, The article review: „ Authentication and market survey of sweet birch (Betula lenta L.) essential oil”. The article presented the the EO extracted from the bark of B. lenta obtained from trusted sources and compares the composition to the oils available in the US market. The EOs of B. lenta were examined to identify chemical markers to help with authentication or adulteration detection. Authors identified: ortho-guaiacol, veratrole, 2-trans-4-cis-decadienal, and 2-trans-4-trans-decadi-enal as natural marker compounds for authentic B. lenta oil. The set goals of the research have been achieved. The presented research results are very interesting and worth publishing after correction of the manuscript. 

Response 1: We appreciate the reviewer’s encouraging, critical and constructive comments on this manuscript. We have taken them fully into account in the revision.  

Comment 2: Title „ Authentication and market survey of sweet birch (Betula lenta L.) essential oil”is adequate to the content. 

Response 2: We appreciate the reviewer’s positive comment. 

Comment 3: The purpose of the research should be better specified. 

Response 3: Thank you for the suggestion. The purpose was to assess the chemical composition of the EO extracted from the bark of B. lenta obtained from trusted sources and to compare the composition with oils available in the US market. 

Comment 4: The same name of the particular EOs components should be use in the text and in the tables; different names are currently provided, for example ortho-guaiacol/o-Guaiacol. 

Response 4: Thank you for the suggestion. It was changed in the text.  

Comment 5: Table 4 should be corrected - values ​​take 2 rows. 

Response 5: Thank you for the observation. The table was set on wrap-text mode.   

Comment 6: In lines 62 and 63 state: “Sweet birch bark essential oils from authentic sources (hydro-distilled in both lab and industrial settings)”. It is not clear; which samples were hydro-distilled in laboratory and which in industrial settings. These are definitely different conditions and a different results. Should be better explained in section Materials and Methods. 

Response 6: Samples A1-A18 were distilled in industrial distillers while A19-A21 were extracted in the laboratory. It was clarified in the text. Results are in table 2. 

Comment 7: SI units should be given. 

Response 7: Thank you for the observation. It was changed in the text. 

Comment 8: The sites of the natural samples of B. lenta should be better characterized. Generally, it is not known where the plants/bark came from. 

Response 8: The plant samples were collected from forest systems in McKean County, Pennsylvania, USA. It was clarified in the text.  

Comment 9: EO yield should be presented for each samples. 

Response 9: Thank you for the suggestion. It was added to the text.  

Comment 10: Literature should be cited after journal requirements (italics of latin names of plant species, unnecessary spaces, etc.). 

Response 10: Thank you for the observation. It was changed in the text.   

Reviewer 4 Report

Dear Authors,

I attach a review of the article „ Authentication and market survey of sweet birch (Betula lenta L.) essential oil”. The presented research results are very interesting from the point of view of  identification of chemical markers which can help in authentication or adulteration detection. In my opinion, the presented research results are very interesting and worth publishing after performing a major corrections.

Remarks

Line 8 - … Sweet Birch (Betula lenta) has several economic and medicinal uses. …

Rev: the economic aspect was omitted in the Introduction

Line 13-15 - … The minor components ortho-guaiacol, veratrole, 2-trans-4-cis-decadienal, and 2-trans-4-trans-decadienal were identified as natural marker compounds for authentic sweet birch oil. …

Rev: the Authors should justify the choice of natural marker compounds; abstract should be corrected

Line 15-16 - … Surprisingly, none of the commercial samples contained any of the identified birch markers. …

Rev: Authors can try to explain this fact

Line 37 - … The tree can reach up to 98 ft tall …

Rev: SI unit should be given

Line 40-45 - B. lenta EO is an anti-inflammatory agent and is suitable for skin problems, rheumatism, and bladder infections. The young sweet birch tree has smooth bark with distinctive horizontal lenticels like other birch species. The mature tree has rough dark brown bark with irregular vertical cracks [8]. Birch bark is one of the abundant, underutilized natural resources [9]. It is the primary waste product of birch processing (about 10-15% of birch mass) [5]. …

Rev: the last sentence suggests that B. lenta is used in production processes; it is true?; whether the literature cited is appropriate?; the text should be corrected.

Line 55-57 - …Therefore, the current study explores the volatile composition of the EO extracted from the bark of B. lenta obtained from trusted sources and compares the composition to the oils available in the US market. …

Rev: “trusted sources” - the origin (habitat) of natural samples is very poorly characterized. The habitat from which the natural samples of birch were collected should be better characterized: plant community, floristic composition of vegetation, soil properties, etc.

Line 57-58 - …To the best of our knowledge, the EO composition of the bark of B. lenta has not been previously reported. …

AND

…. Twenty-seven commercial birch oil samples were obtained from commercial vendors available in the US market. ….

Rev: how it's possible? commercial product has not been previously analyzed?

Line 62-64 - …Sweet birch bark essential oils from authentic sources (hydro-distilled in both lab and industrial settings) and commercial suppliers were obtained from the collections of the Aromatic Plant Research Center (APRC, Lehi, UT, USA). …

Rev: “authentic source” - it is very poor characteristic. The characteristics of the habitat should be presented.

Line 64-65 - … The distillation conditions have been optimized in our previous work on wintergreen essential oil [13]. …

Rev: The distillation conditions in cited work [13] have been optimized for wintergreen, where raw material are leaves. In the presented study the raw material is bark – so there is a fundamental difference. Should be verified.

Line 68-71 - … In the lab, the plant material was distilled for 6-10 hours in a Clevenger-type apparatus. In the industrial setting, the plant material was distilled for 6-10 hours at 6 bar in a 5.5 m3 distiller, starting slowly for 2 hours and then stabilizing around a flow rate of 2L/min. ……

Rev: 2 distillation paths and where are the results?

Line 112 - … of b. lenta bark …

Rev: ?

Line 115 - … inner bark of B. penudula

Rev: ? error in name

Line 146-149 … Interestingly, wintergreen markers such as vitispirane (10 occurrences) and β-dehydroelsholtzia ketone (3 occurrences) were detected in the tested samples. This finding suggests that wintergreen oil is available in the market as birch EO or the addition of wintergreen oil to natural birch oil. …

Rev: this interesting point should be stated in the conclusions

Line 159-160 … Surprisingly, none of the commercial samples contained any of the identified birch markers.

Rev: Authors can try to explain this fact in the conclusions

Line 160-161 … Further investigations on the evaluation of biological activities of B. lenta essential oil are required.

Rev: this statement is somewhat different from the narrative at work

Line 174 - … Table 2. Chemical composition of authentic (A) sweet birch samples

Rev: oil samples were tested, so the Table title should describe it more precisely

Table 2 - A 3

Rev: space remove

Table 2 -

Rev: additionally column with EO yield should be given. Presented range 0.07-1.0% (line 102) is very large.

Line 176 - … Table 4. Chemical composition of commercial (C) sweet birch samples. …

Rev: EOs were analyzed. Therefore, this fact should be taken into account; maybe: commercial birch oil samples?

Line 187 - … ( Betula Papyrifera Marshall) …

Rev: space remove

Line 190-191 - … Trees 11, 190 16–22 (1996). . Trees 1996, 11, 16–22. …

Rev: ?

Line 192-195; 198-199

Rev: literature should be cited as required by the journal

Line 196 - … Betula Lenta …

Rev: italics

Line 201 - … 2021 …

Rev: bold

Line 203 - … Gaultheria Procumbens …

Rev: italics

Line 209 - … Boswellia Dalzielii …

Rev: italics

Line 212 - … Gaultheria Procumbens …

Rev: italics

Line 215-216 - … Betula Nigra …

Rev: italics

Line 219 - … 2004, 2004 …

Rev: ?

Line 227 - … Family. . Pharmaceutical …

Rev: ?

Author Response

Reviewer 4 

Comment 1: Dear Authors, I attach a review of the article „ Authentication and market survey of sweet birch (Betula lenta L.) essential oil”. The presented research results are very interesting from the point of view of  identification of chemical markers which can help in authentication or adulteration detection. In my opinion, the presented research results are very interesting and worth publishing after performing a major corrections. 

Response 1: We appreciate the reviewer’s encouraging, critical and constructive comments on this manuscript. We have taken them fully into account in the revision.  

Comment 2: Line 8 - … Sweet Birch (Betula lenta) has several economic and medicinal uses. …Rev: the economic aspect was omitted in the Introduction 

Response 2:Thank you for your comment. Please refer to line 28 “All parts of the tree, especially the wood, are used as raw materials in the paper and furniture industries in addition to charcoal production, dietary supplements, and cosmetics.” 

Comment 3: Line 13-15 - … The minor components ortho-guaiacol, veratrole, 2-trans-4-cis-decadienal, and 2-trans-4-trans-decadienal were identified as natural marker compounds for authentic sweet birch oil. Rev: the Authors should justify the choice of natural marker compounds; abstract should be corrected 

Response 3: Thank you for the suggestion. The information was added to the text as “The biomarkers are selected based upon three main criteria: (1) the marker should be commercially unavailable or too expensive which renders the adulteration process very costly, (2) The marker should be detected consistently in all the tested authentic EO samples, and (3) A birch EO marker should be found exclusively in birch EO, not in wintergreen and vice versa.” 

Comment 4: Line 15-16 - … Surprisingly, none of the commercial samples contained any of the identified birch markers. …Rev: Authors can try to explain this fact 

Response 4:Thank you for the suggestion. It was clarified in the text.  

Comment 5: Line 37 - … The tree can reach up to 98 ft tall …Rev: SI unit should be given 

Response 5:Thank you for the suggestion. It was changed in the text.  

Comment 6: Line 40-45 - … B. lenta EO is an anti-inflammatory agent and is suitable for skin problems, rheumatism, and bladder infections. The young sweet birch tree has smooth bark with distinctive horizontal lenticels like other birch species. The mature tree has rough dark brown bark with irregular vertical cracks [8]. Birch bark is one of the abundant, underutilized natural resources [9]. It is the primary waste product of birch processing (about 10-15% of birch mass) [5]. …Rev: the last sentence suggests that B. lenta is used in production processes; it is true?; whether the literature cited is appropriate?; the text should be corrected. 

Response 6: Thank you for the observation. That’s correct. All parts of the tree, especially the wood, are used as raw materials in the paper and furniture industries in addition to charcoal production, dietary supplements, and cosmetics. 

Comment 7: Line 55-57 - …Therefore, the current study explores the volatile composition of the EO extracted from the bark of B. lenta obtained from trusted sources and compares the composition to the oils available in the US market. …Rev: “trusted sources” - the origin (habitat) of natural samples is very poorly characterized. The habitat from which the natural samples of birch were collected should be better characterized: plant community, floristic composition of vegetation, soil properties, etc. 

Response 7: The plant samples were collected from forest systems in McKean County, Pennsylvania, USA. It was clarified in the text.  

Comment 8: Line 57-58 - …To the best of our knowledge, the EO composition of the bark of B. lenta has not been previously reported. …AND…. Twenty-seven commercial birch oil samples were obtained from commercial vendors available in the US market. ….Rev: how it's possible? commercial product has not been previously analyzed? 

Response 8: That’s correct. Commercial samples are rarely tested and if tested, the results are not shared with the public, with some exceptions of course.  

Comment 9: Line 62-64 - …Sweet birch bark essential oils from authentic sources (hydro-distilled in both lab and industrial settings) and commercial suppliers were obtained from the collections of the Aromatic Plant Research Center (APRC, Lehi, UT, USA). …Rev: “authentic source” - it is very poor characteristic. The characteristics of the habitat should be presented. 

Response 9: The plant samples were collected from forest systems in McKean County, Pennsylvania, USA. It was clarified in the text.  

Comment 10: Line 64-65 - … The distillation conditions have been optimized in our previous work on wintergreen essential oil [13]. …Rev: The distillation conditions in cited work [13] have been optimized for wintergreen, where raw material are leaves. In the presented study the raw material is bark – so there is a fundamental difference. Should be verified. 

Response 10: Yes, conditions have been checked and verified. 

Comment 11: Line 68-71 - … In the lab, the plant material was distilled for 6-10 hours in a Clevenger-type apparatus. In the industrial setting, the plant material was distilled for 6-10 hours at 6 bar in a 5.5 m3 distiller, starting slowly for 2 hours and then stabilizing around a flow rate of 2L/min. ……Rev: 2 distillation paths and where are the results? 

Response 11: Samples A1-A18 were distilled in industrial distillers while A19-A21 were extracted in the laboratory. It was clarified in the text. Results are in table 2. 

Comment 12: Line 112 - … of b. lenta bark …Rev: ? 

Response 12: Thank you for the suggestion. It was changed in the text.  

Comment 13: Line 115 - … inner bark of B. penudula …Rev: ? error in name 

Response 13: Thank you for the suggestion. It was changed in the text.  

Comment 14: Line 146-149 … Interestingly, wintergreen markers such as vitispirane (10 occurrences) and β-dehydroelsholtzia ketone (3 occurrences) were detected in the tested samples. This finding suggests that wintergreen oil is available in the market as birch EO or the addition of wintergreen oil to natural birch oil. …Rev: this interesting point should be stated in the conclusions 

Response 14: Thank you for the suggestion. It was added to the conclusions.  

Comment 15: Line 159-160 … Surprisingly, none of the commercial samples contained any of the identified birch markers. Rev: Authors can try to explain this fact in the conclusions 

Response 15: Thank you for the suggestion. It was changed in the text.  

Comment 16: Line 160-161 … Further investigations on the evaluation of biological activities of B. lenta essential oil are required. Rev: this statement is somewhat different from the narrative at work 

Response 16: Thank you for the observation. Evaluating the biological activities of B. lenta essential oil is part of our future work. 

Comment 17: Line 174 - … Table 2. Chemical composition of authentic (A) sweet birch samples Rev: oil samples were tested, so the Table title should describe it more precisely 

Response 17: Thank you for the suggestion. It was changed in the text.  

Comment 18: Table 2 - A 3 Rev: space remove 

Response 18: Thank you for the suggestion. It was changed in the text.  

Comment 19: Table 2 - Rev: additionally column with EO yield should be given. Presented range 0.07-1.0% (line 102) is very large. 

Response 19: Thank you for your observation. It was corrected in the test. 

Comment 20: Line 176 - … Table 4. Chemical composition of commercial (C) sweet birch samples. …Rev: EOs were analyzed. Therefore, this fact should be taken into account; maybe: commercial birch oil samples? 

Response 20: Thank you for the suggestion. It was changed in the text.  

Comment 21: Line 187 - … ( Betula Papyrifera Marshall) …Rev: space remove 

Response 21: Thank you for the suggestion. It was changed in the text.  

Comment 22: Line 190-191 - … Trees 11, 190 16–22 (1996). . Trees 1996, 11, 16–22. …Rev: ? 

Response 22: Thank you for the suggestion. It was changed in the text.  

Comment 23: Line 192-195; 198-199 Rev: literature should be cited as required by the journal 

Response 23: Thank you for the suggestion. It was changed in the text.  

Comment 24: Line 196 - … Betula Lenta …Rev: italics 

Response 24: Thank you for the suggestion. It was changed in the text.  

Comment 25: Line 201 - … 2021 …Rev: bold 

Response 25: Thank you for the suggestion. It was changed in the text.  

Comment 26: Line 203 - … Gaultheria Procumbens …Rev: italics 

Response 26: Thank you for the suggestion. It was changed in the text.  

Comment 27: Line 209 - … Boswellia Dalzielii …Rev: italics 

Response 27: Thank you for the suggestion. It was changed in the text.  

Comment 28: Line 212 - … Gaultheria Procumbens …Rev: italics 

Response 28: Thank you for the suggestion. It was changed in the text.  

Comment 29: Line 215-216 - … Betula Nigra …Rev: italics 

Response 29: Thank you for the suggestion. It was changed in the text.  

Comment 30: Line 219 - … 2004, 2004 …Rev: ? 

Response 30: Thank you for the suggestion. It was changed in the text.  

Comment 31: Line 227 - … Family. . Pharmaceutical …Rev: ? 

Response 31: Thank you for the suggestion. It was changed in the text.  

Round 2

Reviewer 1 Report

This revised  version of the original manuscript is a significant improvement, and the efforts of the authors is approciated. 

However, this reviewer is convinced that a precise and accurate determination of minute amounts of tracers ("markers")  would have been preferable, in using GC-MS in SIM mode with internal  standardization, as recommended by IOFI's Working Group of Methods of Analysis, : T. Cachet et al. Flavour Frag. J., 2015, 30 (2), 160-164.

In Table 2 ,  this is a strict requisite. In referring to the IOFI guidelines published in Z. lebensm. Unt. Forsch. , 1991, 192, 530-534, please delete:

the following items:

- (Z)-7-methyl-1,3-dioxaspiro[4.5]decane

- 3-butyl-1,2,4-cyclopentanetrione

- "p-mercapto-Methyl benzoate" (probably wrongly named for methyl p-mercaptobenzoate

Because their biogentic relevance is very low.

Author Response

This revised version of the original manuscript is a significant improvement, and the efforts of the authors is approciated.  

We appreciate the reviewer’s encouraging, critical and constructive comments on this manuscript. We have taken them fully into account in the revision. 

Comment 1: However, this reviewer is convinced that a precise and accurate determination of minute amounts of tracers ("markers") would have been preferable, in using GC-MS in SIM mode with internal standardization, as recommended by IOFI's Working Group of Methods of Analysis, : T. Cachet et al. Flavour Frag. J., 2015, 30 (2), 160-164. In Table 2 ,  this is a strict requisite. In referring to the IOFI guidelines published in Z. lebensm. Unt. Forsch. , 1991, 192, 530-534, please delete: 

the following items: 

- (Z)-7-methyl-1,3-dioxaspiro[4.5]decane 

- 3-butyl-1,2,4-cyclopentanetrione 

- "p-mercapto-Methyl benzoate" (probably wrongly named for methyl p-mercaptobenzoate 

Because their biogentic relevance is very low. 

Response 1: Thank you for the suggestion. They were deleted from the text.  

Reviewer 2 Report

Dear Authors,

The manuscript ID: plants-1852462_v2 entitled „Authentication and market survey of sweet birch (Betula lenta L.) essential oil” has been slightly corrected. The research covers only the composition of the oils. Moreover, most of the oils have not been extracted, but only bought by the Authors. However, the EO composition of B. lenta bark has not been described in such detail before and according to the Authors, this is the first report that has identified birch biomarkers. I have no other comments on this article.

With highest regards,

Author Response

Comment 1: 

Dear Authors, 

The manuscript ID: plants-1852462_v2 entitled „Authentication and market survey of sweet birch (Betula lenta L.) essential oil” has been slightly corrected. The research covers only the composition of the oils. Moreover, most of the oils have not been extracted, but only bought by the Authors. However, the EO composition of B. lenta bark has not been described in such detail before and according to the Authors, this is the first report that has identified birch biomarkers. I have no other comments on this article. 

With highest regards, 

Response 1:We appreciate the reviewer’s encouraging, critical and constructive comments on this manuscript. We have taken them fully into account in the revision. The biological activity of B. lenta L. essential oil is part of our future work. 

Reviewer 4 Report

Acceptance after applying a correction

Line 55-56: … of wintergreen (Gaultheria spp.) …

Rev: spp. no italics

Line 77-80: … red maple (Acer rubrum L.), sugar maple (Acer saccharum Marshall), black cherry (Prunus serotina Ehrh), American beech (Fagus grandifolia Ehrh), Eastern hemlock (Tsuga canadensis (L.) Carriere), and yellow birch (Betula alleghaniensis Britt). …

Rev: English name; english name; big and small letters; check the spelling of the species' names.

Table 4 …

Rev: correction required: column 3

Line 248: … Gaultheria procumbens L . and Betula Lenta L . …

Rev: should be: Gaultheria Procumbens L. and Betula Lenta L. or Gaultheria procumbens L. and Betula lenta L. – one convention must be applied

Rev:  the same principle should be applied to all names in the literature section

Line 276: … Cortex Betula …

Rev: this is a latin name, maybe in italics?

Author Response

Reviewer 4 

Acceptance after applying a correction  

We appreciate the reviewer’s encouraging, critical and constructive comments on this manuscript. We have taken them fully into account in the revision.  

Comment 1: Line 55-56: … of wintergreen (Gaultheria spp.) … 

Rev: spp. no italics 

Response 1: Thank you for the suggestion. It was corrected in the text.  

Comment 2: Line 77-80: … red maple (Acer rubrum L.), sugar maple (Acer saccharum Marshall), black cherry (Prunus serotina Ehrh), American beech (Fagusgrandifolia Ehrh), Eastern hemlock (Tsuga canadensis (L.) Carriere), and yellow birch (Betula alleghaniensis Britt). …Rev: English name; english name; big and small letters; check the spelling of the species' names. 

Response 2: Thank you for the suggestion. The spelling and names were double-checked.  

Comment 3: Table 4 …Rev: correction required: column 3 

Response 3: Thank you for the suggestion. The table was readjusted to space.   

Comment 4: Line 248: … Gaultheria procumbens L . and Betula Lenta L . … 

Rev: should be: Gaultheria Procumbens L. and Betula Lenta L. or Gaultheria procumbens L. and Betula lenta L. – one convention must be applied 

Rev:  the same principle should be applied to all names in the literature section 

Response 4: Thank you for the suggestion. All the botanical names were rechecked.   

Comment 5: Line 276: … Cortex Betula …Rev: this is a latin name, maybe in italics? 

Response 5: Thank you for the suggestion. All the botanical names were rechecked.   

Comment 6: Line 8 - … Sweet Birch (Betula lenta) has several economic and medicinal uses. …Rev: the economic aspect was omitted in the Introduction 

Response 6: Thank you for the suggestion. Please refer to lines 30-32: All parts of the tree, especially the wood, are used as raw materials in the paper and furniture industries in addition to charcoal production, dietary supplements, and cosmetics [5,6]. 

Comment 7: Line 13-15 - … The minor components ortho-guaiacol, veratrole, 2-trans-4-cis-decadienal, and 2-trans-4-trans-decadienal were identified as natural marker compounds for authentic sweet birch oil. Rev: the Authors should justify the choice of natural marker compounds; abstract should be corrected 

Response 7: Thank you for the suggestion. It was changed in the text.